# Impact of Motile Ciliopathies on Human Development and Clinical Consequences in the Newborn

**DOI:** 10.3390/cells11010125

**Published:** 2021-12-31

**Authors:** Rachael M. Hyland, Steven L. Brody

**Affiliations:** 1Department of Pediatrics, Division of Newborn Medicine, Washington University in Saint Louis School of Medicine, Saint Louis, MO 63110,USA; rmfewell@wustl.edu; 2Department of Medicine, Division of Pulmonary and Critical Care Medicine, Washington University in Saint Louis School of Medicine, Saint Louis, MO 63110, USA

**Keywords:** motile cilia, human development, primary ciliary dyskinesia, neonate

## Abstract

Motile cilia are hairlike organelles that project outward from a tissue-restricted subset of cells to direct fluid flow. During human development motile cilia guide determination of the left-right axis in the embryo, and in the fetal and neonatal periods they have essential roles in airway clearance in the respiratory tract and regulating cerebral spinal fluid flow in the brain. Dysregulation of motile cilia is best understood through the lens of the genetic disorder primary ciliary dyskinesia (PCD). PCD encompasses all genetic motile ciliopathies resulting from over 60 known genetic mutations and has a unique but often underrecognized neonatal presentation. Neonatal respiratory distress is now known to occur in the majority of patients with PCD, laterality defects are common, and very rarely brain ventricle enlargement occurs. The developmental function of motile cilia and the effect and pathophysiology of motile ciliopathies are incompletely understood in humans. In this review, we will examine the current understanding of the role of motile cilia in human development and clinical considerations when assessing the newborn for suspected motile ciliopathies.

## 1. Introduction

### 1.1. Primary vs. Motile Ciliopathies and Human Development

Human genetics has revealed cilia as essential directors of human development [1,2]. Cilia, the complex hair-like organelles that extend from the surface of cells, are found throughout the body. Two classes of cilia, primary and motile, share basic structures but have distinct components, cellular distribution, and functions that influence development. Primary (sensory) cilia are well known for their role in cell signaling during development [3,4]. These are present on almost every cell type as sensing and signal transducing organelles [1,4]. Genetic defects in primary cilia encompass syndromes involving multiple organ systems, most frequently affecting the structure of the kidney, liver, and pancreas, but also causing skeletal anomalies, sensory organ deficits, and brain malformations [2]. In contrast, motile cilia are liquid-moving organelles. They contain specialized motor protein complexes that include members of the dynein family to facilitate bending motions that move fluids. Motile cilia are limited to a few cell types: nodal cells of the embryonic node during development, epithelial cells lining the respiratory tract, brain ventricles, and fallopian ducts, and the sperm flagellum. Despite shared structures and appearance, developmental syndromes affecting the two cilia types are distinct, as motile ciliopathies are nearly exclusively the result of genetic variants that hinder their motor proteins [5]. The function and organization of motile cilia during human development is incompletely understood. Here, we summarize knowledge of motile cilia specifically in human development and the impact of motile ciliopathies in the newborn.

### 1.2. Motile Cilia Structure

Motile and primary cilia both arise from an anchoring centriole as a scaffold composed of 9 microtubule doublets (the axoneme), as recently reviewed [3,6,7]. Each motile cilium additionally has tens of thousands of motor proteins and regulatory proteins, which when altered or absent result in PCD [5,8,9]. Large motor protein complexes (dyneins) are docked at ruler-like intervals along each of the microtubule doublets of the axoneme and are named for their relative location on the scaffold as inner or outer dynein arms (IDA, ODA) [10]. Each 96-nm repeated segment along the axoneme contains four ODAs and seven IDAs [11]. Motile cilia also have two additional single microtubules in the middle of the ring of nine, called the central pair or central apparatus, which is linked to the outer nine microtubule doublets (Figure 1). This classic “9 + 2” arrangement found in cilia of the airways, brain ventricles, fallopian tube, and sperm facilitates the wave-like beating [12,13]. An exception is the motile cilia of the embryonic node, which lack the central pair, resulting in a rotational rather than bending motion [14,15].

### 1.3. Appearance of Motile Cilia in the Human Embryo

Motile cilia appear in three structures during development which significantly impact the developing human fetus and newborn: the embryonic node, respiratory tract, and brain epithelium (Figure 1). Motile cilia first arise in the embryonic node, a transient structure present around three weeks of gestation in the human embryo [16,17]. Motile cilia next appear during fetal organogenesis within the developing respiratory tract, in the trachea at 7 weeks gestation and within the intrapulmonary airways at about 10 weeks, commencing with the early stages of lung development [18,19]. In the developing human brain, motile cilia on ependymal cells lining the ventricles emerge at around 25 weeks [20,21,22]. Of unknown function, motile cilia are present in the fetal esophagus at around week 16 of gestation but disappear by birth [23,24]. Multiciliated cells lining the fallopian tubes in females appear by approximately 14 weeks, but their known function is much later, establishing the fluid flow that carries oocytes [25,26,27,28]. The male sperm flagellum shares components with motile cilia, but only appear during spermatogenesis after the onset of puberty [29,30,31].

## 2. Motile Cilia Disease in the Human: Primary Ciliary Dyskinesia (PCD)

Failure of motile cilia function is categorized as a single genetic multisystem disorder called primary ciliary dyskinesia (PCD), which affects approximately 1 in 10,000 births [32,33]. PCD is genetically heterogeneous, with pathologic variants in over 60 different genes coding for motor protein assembly, structure or regulation causative of disease [33]. Classic features of PCD as described in children and adults are persistent wet cough, nasal congestion, chronic sinopulmonary infection, chronic otitis media, organ laterality defects, bronchiectasis, and infertility. Kartagener’s syndrome is a prior-used term for a subgroup of PCD with situs inversus, sinusitis and bronchiectasis. The neonatal presentation of PCD is tied to pathology from motile cilia defects during development and at birth. Clinical features include neonatal respiratory distress in >80% of newborns, laterality defects in approximately 50% (with laterality-based developmental cardiac defects as a cause of congenital heart disease in a small percent), and in the exceptionally rare case (<1%) hydrocephalus (Table 1) [33].

At least 22 of the 60-plus known PCD genes have been specifically reported to cause neonatal symptoms (Table 2), though it is likely that pathogenic variants in all PCD genes could impact the neonate. Comprehensive frequencies of most of these variants are unavailable, many are limited to case reports. Motile cilia disorders in PCD are nearly always the result of autosomal recessive genetic variants, with the exception of a few X-linked recessive genes (*PIH1D3*, *RPGR*, *OFD1*) and one autosomal dominant gene (*FOXJ1*) [33]. Pathogenic variants in components of motor protein structure result in absent or ineffective cilia beating, the most common reported in the neonate are mutations in ODA proteins (e.g., *DNAH5*, *DNAH11*, *DNAI1*) or in ‘ruler’ complexes (*CCDC39*, *CCDC40*) which determine placement of IDA and regulatory proteins. Pathogenic variants in genes that are major controllers of ciliogenesis (*MCIDAS*, *FOXJ1*) can result in decreased generation of motile cilia and a PCD-like phenotype in neonates [34]. Below, we review the developmental features of motile ciliogenesis in specialized tissues and the relationship to features of neonates with PCD.

## 3. Motile Cilia in the Embryonic Node during Human Embryogenesis

### 3.1. Development of Embryonic Node Motile Cilia

The mechanism of body asymmetry patterning was elucidated in the 1990′s when cilia function in the embryonic node was first proven to be required for left-right determination in mouse models [14,56,57]. The embryonic node, also referred to as the primitive node or the left-right organizer, is a transient structure that arises from the primitive streak at gastrulation, likely during the third week of human embryogenesis [17]. Data identifying the precise timing in human embryos are limited, to our knowledge scanning electron microscopy of the human node have not been published. Observations from non-human models have shown that as the streak elongates, a central ‘pit’ of endodermal cells form the embryonic node [57]. At the midline and extending outward from the pit are 200–300 motile monociliated nodal cells [58]. Analysis of mouse and zebrafish models show two different populations of monociliated cells within the node, motile 9 + 0 centrally and sensory 9 + 0 at the lateral edges [59,60,61]. High-speed video microscopy of the mouse node shows nodal cilia with clockwise rotational movement at 10 Hz which generates a leftward fluid flow across the node [15]. This is presumed to be from the predominant population of 9 + 0 motile nodal cilia, with rotational as opposed to bending motion owing to the lack of a central pair [14,15,62]. Cilia with varied ultrastructure have also been identified within the node. Cilia with 9 + 2 and 9 + 4 structures have been found in the zebrafish, mouse, and rabbit embryonic node but their function and movement pattern are not yet known; it is presumed they would beat rather than spin (due to the presence of central microtubules) [63,64,65,66,67]. Motility requires transcription factor FOXJ1, among others, and functional IDA and ODA motor complexes [68]. The critical period of motile cilia-related gene expression and nodal flow is transient, over several hours in the mouse, before the node is absent at the six-somite stage of embryogenesis (approximately 28 days of human gestation) [69,70]. A cascade of asymmetric gene expression ensues, featuring *NODAL*, *LEFTY* and *PITX1* on the left side of the embryo, resulting in activation of tissue-specific programs for asymmetric organogenesis [71,72].

The cilia-generated leftward laminar flow of extraembryonic fluid within the node cavity is essential for left-right asymmetry determination. Fluid dynamic models have shown that the leftward flow pattern is directed by a posterior tilt of the nodal cell and cilia axis, which is postulated to be driven by WNT5-mediated signaling for planar cell polarity [73,74,75]. The mechanism for translation of nodal flow to left-right asymmetry, extensively studied in genetic mouse and zebrafish models, remains unresolved. Predominant theories are that either the generated fluid flow within the node causes leftward movement of signaling molecules within the nodal pit to initiate a downstream program (morphogen gradient model) or that leftward flow generates mechanical stress resulting in the influx of calcium in an asymmetric cluster of surrounding sensory cilia, triggering the molecular program (mechanosensory, or ‘two cilia’, model) [60,62,70].

### 3.2. Left-Right Body Asymmetry

The ultimate development of normal body asymmetry (situs solitus, SS) is identified by the relative positions of the visceral organs and their blood supply; the heart on the left side of the chest, lungs with three lobes on the right and two on the left, liver on right, spleen and stomach on the left, and asymmetric gut rotation. Complete reversal of this alignment is called situs inversus totalis (SI), which is the most common clinical presentation of a laterality defect in PCD and may be asymptomatic [60,76]. Intermediate laterality configurations (situs ambiguous, SA, or heterotaxy) are a diverse spectrum of anatomic variants above and below the diaphragm, often associated with congenital heart disease or splenic abnormalities. In some reviews, laterality defects with complex cardiac malformations differentiate heterotaxy from non-cardiac SA, though there is not a known unique developmental pathogenesis [77].

### 3.3. PCD Mutations Associated with Laterality Defects

Pathogenic variants in genes coding for ciliary motor components, including motor assembly proteins and ODA motor proteins, result in node cilia failure and random laterality generation (SS, SI, or SA) [33,78]. In contrast, genes coding for proteins that are components of the radial spoke proteins or central pair complex do not have laterality defects. Mutations in several of these central pair and radial spoke ciliary structures have been shown to result in an abnormal residual twirling motion of cilia in airway multiciliated cells, similar to the normal motion of the embryonic node 9 + 0 cilia, likely allowing sufficient nodal function for establishment of normal LR asymmetry [50,79]. Phenotypes of individuals with mutations in multiciliogenesis genes are divergent based on their function; most, but not all, classes of mutations affect the function of monociliated nodal cilia, impacting the determination of laterality. Of the multiciliogenesis genes, *CCNO* and *MCIDAS* are not associated with laterality defects but *FOXJ1* is. *CCNO* is a member of the centriole amplification pathway, which is not required in the monociliated node cells. Why *MCIDAS* variants are not associated with laterality defects is unknown. *FOXJ1* variants are associated with laterality defects including SI owing to the fundamental upstream role of *FOXJ1* in inducing all forms of motile cilia expression, including in the node [53].

### 3.4. Clinical Features of Laterality Defects in Neonates with PCD

Laterality defects are manifest by SI in nearly half of neonates with PCD while another 10–12% fall along the spectrum of SA [80]. Population-based studies suggest SI incidence in all newborns is 1 in 6–8000 (higher than the frequency of PCD) reinforcing that there are factors downstream of nodal cilia that when altered may cause SI [76,81,82]. Patients with isolated SI occurring with normal cilia function are generally not clinically affected. However, heterotaxy (SA) is associated with increased morbidity secondary to complex congenital heart disease and polysplenia (multiple splenic tissues) or asplenia (lack of spleen) known to cause immune defects with risk for infection [80,83,84]. The presence of SI or SA should be considered an early warning bell for clinicians and will typically trigger a genetic workup to an earlier diagnosis. A recent cohort study of pediatric PCD patients showed that those with laterality defects were more likely to be diagnosed in infancy. Overall, 9.4% were diagnosed in the first two months of life and of these 78% had a laterality defect, while the entire study population had a typical PCD incidence of 48% laterality defect [85].

### 3.5. Relationship of Motile Cilia and Laterality Determination with Cardiac Development

Clinically, there has been a long-standing known association between PCD and congenital heart disease (CHD) [80,86]. The genetic relationship was established in murine models with variants in genes known to cause PCD and immotile nodal cilia [9,87,88]. Population-based studies of PCD cohorts show increased association with cardiac disease in those with ODA and IDA gene mutations as opposed to central pair or radial spoke defects [86]. The vast majority of CHD in PCD patients (not including complete dextrocardia) is in the setting of situs ambiguous, affecting up to 3–6% [78,80,86]. Cardiac defects associated with heterotaxy include a spectrum from atrial isomerism to complex single ventricle or great vessel transposition, the latter carrying significantly higher morbidity [77,80,84].

## 4. Motile Cilia in the Developing Human Respiratory Tract

### 4.1. Motile Cilia in the Lung Airway Epithelium

Multiciliated cells are present in the upper and lower airway on the epithelium of the paranasal sinuses, the Eustachian tubes of the ears, the adenoid glands, as well as throughout the lung’s conducting airway. Airway multiciliated cells contain 200–300 motile (9 + 2) cilia per cell that beat in synchrony, and, as proven by PCD, are essential for airway clearance [24,89]. Mucus produced by secretory cells traps particles and bacteria, while neighboring multiciliated cells facilitate upward fluid propulsion to push debris-laden mucus out of the airway. Human airway motile cilia are reported to beat at a frequency of roughly 12 Hz. Beat frequency is reported to be slightly faster in neonates (12–13 Hz) compared to adults (10–12 Hz), although many factors can affect the ex vivo measurement of cilia beat frequency measurement including temperature and calcium concentration [90,91].

### 4.2. Development of the Motile Ciliated Airway Epithelium

Motile cilia arise early in the fetal airway during the pseudoglandular stage of lung development [19]. Lung development proceeds through five canonical stages based on anatomic features; embryonic, pseudoglandular, canalicular, saccular, and alveolar, which have some inherent overlap as the lung develops proximally to distally via branching morphogenesis. The fetal, or embryonic stage comprises early lung organogenesis commenced by the outpouching of the foregut forming a primitive trachea, followed by lung buds and bronchi then visceral and parietal pleura, at 4–7 weeks [92,93]. Respiratory epithelium in the embryonic lung is primarily undifferentiated, expressing transcription factor SOX2 [94,95,96]. Motile ciliated cell differentiation begins in the pseudoglandular stage (roughly weeks 5–17) and continues through the canalicular (weeks 16–26), and saccular (weeks 24–38) stages. Molecular markers of ciliogenesis throughout these stages are better described in mice than in human tissues [97,98]. The airway epithelium is columnar proximally and lowers in height towards cuboidal-shaped cells distally with growth of new bronchioles during the pseudoglandular phase. In this stage, differentiation proceeds proximal to distal, at which time SOX2+ cells become TP63 basal cells that give rise to secretory and ciliated cells [94,95]. NOTCH signaling is key in regulation of this differentiation. Activation of NOTCH leads to the secretory fate, while inhibition of Notch defaults to differentiation of basal cells into multiciliated cells [99,100].

The future multiciliated airway cell is marked by the presence of a single primary cilium with a parent (or, ‘mother’) centriole, which are not found on secretory cells [95]. Activation of transcription factors GEMC1, E2F4 and E2F5, halt cell cycle progression and form a complex with transcriptional cofactor MCIDAS [101,102]. Under the influence of MCIDAS and cell cycle proteins (e.g., CCNO), centriole numbers are amplified by two pathways: replication of centrioles from procentrioles and de novo by formation of structures called deuterosomes [89,103,104]. Deuterosomes release new centrioles that traffic to the cell surface as basal bodies, providing a template that gives rise to the cilia. Each basal body has a foot process for anchoring and coordinating unidirectional beating of the cilia. A newly identified single ‘hybrid’ cilium in each cell, named for its shared features between primary and motile cilia, is required for basal body alignment and coordinated cilia beating in the airway [105]. GEMC1 and MCIDAS transcriptional cofactors induce ciliogenesis by activating transcription factors FOXJ1, FOXN4, and TP73 which program the production of components required for motile cilia including the motor protein complexes [104,106,107]. Assembly of the motor protein-containing IDA and ODA complexes occur within the cell cytoplasm via a poorly understood process. Current concepts suggest that a set of dynein axonemal assembly factors (DNAAFs) including DNAAF2, HEATR2, ZMYND10, SPAG1, LRRC6 and others are required for assembly of the dynein motor containing complexes that form the IDA and ODA. Mutation of any one of the DNAAFs leads to PCD [18,108].

### 4.3. Genetic Features of Airway Motile Cilia Defects in Neonates with PCD

As is apparent from molecular roles of regulator proteins described above, impaired airway epithelial motile ciliary function can result from mutations in genes coding for proteins along the pathway of motile ciliogenesis (Table 2), leading to impaired airway clearance. Respiratory distress in the neonate is a recently identified hallmark of PCD, estimated to occur in 60–85% of patients, but is under appreciated [37,109,110,111]. Pathogenic variants in the genes that control key stages of multiciliogenesis, *MCIDAS*, *FOXJ1*, *CCNO*, and *TP73* result in sparse motile cilia and impaired clearance [95]. However, ciliary axoneme generation itself is normal in most of the genetic causes of PCD, indicating that normal motor function is a predictor of airway clearance. As such, the most common PCD mutations are also those that result in respiratory symptoms at birth, especially when pathogenic variants affect motor protein genes coding for the ODA: *DNAH5*, *DNAH11*, and *DNAI1* [37]. Additional groups of PCD gene mutations causing respiratory distress are the dynein assembly proteins (e.g., *DNAAF2*, *DNAAF5*) and the ruler proteins (*CCDC39*, *CCDC40*). Mutations in ruler protein genes generally leads to neonatal respiratory distress and worse respiratory function in childhood and adulthood, compared to isolated ODA defects [112].

### 4.4. Clinical Features of Airway Motile Cilia Defects in Neonates with PCD

Impaired mucociliary clearance due to a motile ciliopathy is thought to allow mucus and particulate matter to obstruct the distal airways at birth during the transition from liquid to air breathing. The result is decreased lung inflation and atelectasis (collapse), neonatal respiratory distress, and a predisposition to respiratory infections [37,113,114,115]. Despite the high frequency of respiratory distress within the PCD population, the likelihood of an individual newborn presenting with respiratory distress secondary to PCD is low. Term neonatal respiratory failure, irrespective of underlying diagnoses, occurs in around 2 cases per 1000 live births, typically related to disorders of transition, or infectious or inflammatory conditions (sepsis, meconium aspiration, birth asphyxia); all disorders which are primarily considered to affect the alveolus, not the conducting airway [116,117,118]. Accordingly, the majority of focus in neonates historically has been in alveolar disease; PCD reports have been confined to case series, and standard reviews of neonatal respiratory distress do not reference ciliary disorders [111,119]. This gap highlights that respiratory distress is common, PCD is rare, and clinicians must be vigilant to recognize the disorder in the neonatal period.

Case series to date indicate that the neonatal PCD respiratory phenotype has several distinguishing features (Figure 2). First, PCD is not associated with prematurity, so unlike the majority of newborns with respiratory distress, babies with PCD are typically full-term. Second, newborns with PCD typically develop respiratory distress accompanied by lobar collapse (atelectasis) at around 12–24 h after birth. The timing of onset and presence of atelectasis differs clinically from the more common respiratory distress syndromes affecting term newborns (e.g., transient tachypnea of the newborn), which generally present within two hours of birth [111,114]. Third, supplemental oxygen for persistent hypoxemia is often required for more than 48 hours, commonly days to weeks [37,120]. Finally, in the neonatal PCD phenotype there is often a period of symptomatic, with most newborns experiencing a temporary resolution of symptoms until later infancy when classic re-presentation includes chronic rhinosinusitis, wet cough, and recurrent otitis media [109,113,114].

### 4.5. Acquired Ciliary Dysfunction in the Newborn

Aberrant ciliary function from disrupted development or direct injury to the conducting airway may present with respiratory symptoms similar to genetic ciliopathy (PCD) in the neonate, but an acquired ciliopathy is not well defined. The majority of studies of neonatal airway injury are in animal models with mechanical ventilation, which consistently show airway inflammation and injury to ciliated cells [121,122]. Limited histologic studies in premature infants who received mechanical ventilation have shown severe epithelial denudation with ciliary anomalies accompanied by significantly decreased numbers of mucous cells, supporting the concept of acquired ciliopathy, as opposed to increased secretion of mucus causing pathology [123]. Some airway pathogens including *Pseudomonas aeruginosa* and respiratory syncytial virus have virulence mechanisms that impair cilia function, by means of bacterial cell toxins such as pyocyanin or viral suppression of ciliogenesis transcription factor expression [124,125,126]. PCD-based therapeutic avenues focused on ciliary augmentation or enhanced mucus clearance strategies would likely translate well to acquired cilia defects.

## 5. Motile Cilia in the Developing Human Brain Ventricular System

### 5.1. Motile Cilia in the Brain Ventricular Epithelium and Cerebral Spinal Fluid Flow

Motile cilia are found in the ependyma and choroid plexus of the brain ventricular system and the spinal cord central canal, where they may have a faciliatory role in cerebral spinal fluid (CSF) flow [16,127,128]. CSF is secreted by a specialized subpopulation of epithelial cells of the choroid plexus and flows through the ventricles to the subarachnoid space, as a conduit for cell communication and homeostasis [129]. CSF is ultimately absorbed into the venous system through arachnoid granulations, the projected folds of the arachnoid matter [129]. Ependymal cell motile cilia beat frequency has not been established in humans but in mice and rats they roughly double that of respiratory cilia, although with regional variability [130,131].

### 5.2. Developmental Control of Motile Ciliogenesis in the Brain Ventricular Ependymal Cells

The ventricular system emerges soon after completion of neural tube closure at human embryonic day 30, as cells induce radial glial cell progenitors under control of *FOXG1*, *LHX2*, *PAX6*, and *EMX2*, as shown in mouse models [132,133]. Further differentiation starts around week 7 of human development in a caudal-to-rostral direction in the brain and rostral-to-caudal in the spinal cord [134]. Radial glial cells are the stem cells that give rise to the multiciliated choroid and ependymal cells well after the ventricles are formed, at around 25 weeks’ gestation [133,134]. Ependymal cells ciliate soon after, but cilia are immature in orientation and short in length until near term [127,134,135]. Detailed analysis of cilia throughout the ventricular system of the mouse brain has identified at least three types of ciliated cells that control CSF flow on the ependymal wall: those with abundant cilia ranging 20–100 per cell, some with one or two motile cilia found in the third and fourth ventricle, and the non-motile monociliated cells in the third ventricle [136,137,138]. The regulation of differentiation of ciliated ependymal cells in the human is presumed to be similar to animal models; dependent on NOTCH signaling with *GEMC1* and *MCIDAS* as upstream regulators coupled with downstream FOXJ1 expression for generation of motile cilia [24,139,140,141]. As in the respiratory tract, planar cell polarity protein Vangl2 is implicated for ciliary beat alignment [142,143].

### 5.3. Clinical Features of Brain Ependymal Motile Cilia Dysfunction in Neonates with PCD

Nearly all genetic mouse models of PCD develop hydrocephalus, consistent with the function of motile cilia to move fluids (CSF), but in humans with PCD hydrocephalus is exceedingly rare at less than 1% of the overall PCD population [9,16,144,145,146]. Multiple theories have been advanced to explain the relative rarity in PCD patients. First, the ventricles are anatomically different in rodents and humans. Humans have a larger ventricular system with relatively wider connecting passages, less prone to obstruction [16]. Second, ependymal cells may have additional roles in the mouse compared to the human brain, including facilitating production of CSF. Third, the glycoprotein composition of CSF is different between species, perhaps affecting flow [144]. Additionally, there are differences in brain development, notably the timing of forebrain neurogenesis [140].

In all human infants, hydrocephalus has an incidence of 0.5–1 cases per 1000 births, and is associated with developmental delay, cerebral palsy, and seizure disorders [147]. Untreated hydrocephalus carries a mortality rate of 50% [147,148]. While it may arise from any defect related to CSF production and flow, the most common etiologies include congenital infection of the newborn, developmental anomalies obstructing CSF flow (e.g., aqueductal stenosis), or intraventricular hemorrhage [149].

When reported, hydrocephalus in PCD patients is most closely associated with variants in genes resulting in reduced generation of motile cilia (*FOXJ1*, *MCIDAS*, *TP73*, *CCNO*). In patients with these pathogenic variants the prevalence of hydrocephalus approaches 10% [150,151]. Variants in *FOXJ1* are most consistently associated with hydrocephalus, with one case series present in all of 11 subjects [53]. In mouse models, *FOXJ1* has a key role in neurogenesis and ependymal cell maturation, the alteration of which may contribute to hydrocephalus [141]. Other reported cortical malformations are lissencephaly in patients with *TP73* mutations and abnormal respiratory cilia function [151]. Mild ventricular dilation and severe prenatal hydrocephalus has been identified only sporadically in case reports of PCD families with gene mutations affecting structural defects in central microtubules and dynein arms [145,152]. Comprehensive evaluation of ventricular size has not been performed in cohorts of patients with PCD. It is possible that subclinical flow-related changes are more often present or that mild ventriculomegaly occurs short of macrocephaly, with undetermined consequence. A small cohort study of PCD children revealed increased behavioral and developmental delay compared to control groups [153]. It is possible that subtle neurologic changes as the result of altered brain motile cilia contribute to these developmental differences.

## 6. Evaluation and Management of the Newborn with Suspected PCD

### 6.1. Identification of the Newborn with Suspected PCD

Overall, PCD is considered to be under-recognized, with estimates suggesting that as few as 10% of all individuals with PCD have been definitively diagnosed and appropriately managed in specialized PCD centers [154]. Diagnosis of PCD in the newborn period is particularly challenging and requires a high index of suspicion. In a recent retrospective cohort study of pediatric patients with PCD only 9% were diagnosed in infancy (in this cohort, 55% had neonatal respiratory distress and 42% laterality defects, with co-occurrence in 29%) [85]. PCD diagnostic guidelines by the European Respiratory Society (ERS) and the American Thoracic Society (ATS) are both derived from data in children with the mean ages of 7–9 years and predicated on historical features [110,155]. The ATS diagnostic algorithm for PCD requires two of four key clinical features: unexplained neonatal respiratory distress in a term infant, year-round daily cough beginning before 6 months of age, year-round daily nasal congestion beginning before 6 months of age, and organ laterality defect. By definition, two of these criteria require progression past infancy. Similarly, the ERS algorithm commences by asking if a daily wet cough is present from early childhood, not applicable to the neonatal population.

Diagnosis in the neonatal period requires recognition of the clinical phenotype (Figure 2). Evolving experience and data suggest that evaluation for PCD in the neonatal period is based on a phenotype that varies from the classical features identified an older child. Instead, the diagnosis is centered on respiratory distress, and co-occurring laterality defects or a family history of PCD (Table 3).

### 6.2. Diagnostic Testing of the Neonate with Suspected PCD

Assessment of the neonate is centered in imaging and genetic testing (Table 4). Chest X-ray will evaluate for atelectasis and screen for laterality defects [37,120]. Chest CT and surgical lung biopsy are not indicated in the neonate to evaluate for PCD but are part of diagnostic algorithms for interstitial lung diseases that cause respiratory distress in the term newborn and should be considered only in consultation with subspecialists [156]. Echocardiography should be performed to evaluate for congenital heart disease and related laterality defects of the vessels. To assess for visceral organ laterality defects, abdominal ultrasound with attention to polysplenia or asplenia is necessary. Simple measurement of the occipital frontal head circumference standardized appropriately against gestational age-adjusted growth curves will assess for macrocephaly and screen for hydrocephalus. As ventricular dilatation is likely under-recognized in PCD, screening head ultrasound is a reasonable consideration. Genetic testing using commercially available PCD panels is the most straightforward confirmatory approach in the newborn. Standard panels now include over 32 genes, and these extended spectrum panels have excellent specificity and sensitivity generally around 70% depending on the panel [8,120,157]. Testing should be done in conjunction with genetic or PCD specialists to assess specific allelic mutations and appropriately interpret variants of unknown significance.

Nasal nitric oxide (nNO) testing is a useful screening test in older children and adults with suspected PCD [158,159]. Testing is not validated in children <5 years of age or recommended in the neonate [110,120]. Pilot studies of nNO measurement in newborns have shown feasibility, though values are significantly lower when compared to control or disease reference ranges. Low levels are attributed to physiologic differences in nNO production in the underdeveloped sinuses of the infant [160]. Evaluation of cilia ultrastructure by transmission electron microscopy (TEM), or cilia movement by high-speed video light-microscopy (HSVM) are technically difficult and should be limited to specialized PCD centers [110,120]. Obtaining airway cells in the neonate poses technical challenges. Brush biopsies of the nares or upper respiratory tract require significant expertise to isolate multiciliated cells for TEM, HSVM, or culture.

### 6.3. Management of the Newborn with Suspected or Confirmed PCD

There are no specific treatments for PCD. Instead, PCD is managed with symptom-directed care based in expert opinion and extrapolation from other disease processes that have obstructive physiology such as cystic fibrosis [120,161]. Likewise, treatments for adults and children are extrapolated to address problems in the newborn.

In the neonatal period, management of respiratory distress often requires positive pressure ventilation with or without supplemental oxygen [37]. Airway clearance is the cornerstone of PCD treatment but is more difficult in the newborn who cannot actively participate in clearance maneuvers. Daily inhaled normal saline and chest physiotherapy with gentle manual percussion are recommended [120]. Data on drug therapy for PCD remain nearly all anecdotal. Mucolytics (such as n-acetylcysteine or dornase alpha) co-opted from their use in CF patients have not been shown to be beneficial and are not recommended [120]. Case reports have suggested regular use of nebulized dornase alpha improve PCD patient pulmonary function testing and pulmonary symptoms, however benefits are unlikely in the neonate where high neutrophil burdens (associated with high airway DNA levels) are not established by chronic infection [162]. Likewise, inhaled corticosteroids are not advised in the neonatal period due to insufficient evidence of benefit and some evidence for increased associated mortality with use [163,164,165]. Beta-adrenergic agonists, sometimes used as bronchodilators in PCD, increase cilia beat frequency and theoretically may have a role in improving residual cilia function, though no current data support use for this purpose [164,166]. Modulating calcium-dependent cilia activity using phosphodiesterase inhibitors to increase cilia beating remains theoretical [167].

Specialty referral is often a component of care for infants with PCD. For those neonates with co-occurring laterality defects, particularly with congenital heart disease and heterotaxy syndromes, consultation with a cardiologist is paramount. Isolated laterality defects, not involving the heart or splenic functional anomalies, do not require further management, as they have been not been found to be associated with negative outcomes. In the case of splenic abnormalities referral should be made to an immunologist as functional deficits frequently accompany this disorder [83]. The rare occurrence of ventricular enlargement or hydrocephalus requires neurologic subspecialty care [168]. Genetic counseling should be offered if the diagnosis of PCD is confirmed.

Neonates with PCD should be considered candidates for RSV immune prophylaxis during the first two years of life [120,169]. Prophylaxis with antibiotics is not recommended, although a large multicenter randomized trial in preventive daily azithromycin use in PCD patients ages 7–50 reduced respiratory exacerbations, infants were not studied [170]. PCD patients are at high risk of hearing loss, which is reported in up to half of all PCD patients. Screening hearing exams should be obtained in all neonates at the time of diagnosis, and routine audiology follow-up continued under the guidance of otolaryngology consultants [171].

PCD management in the neonatal period is only the beginning of a life-long journey for patients. Care should be directed by PCD centers with routine follow-up, including management of potential infertility. Studies on the impact of disease on quality of life unfortunately show significant burden to affected individuals, which may be partially mitigated by early care [172]. Therapies to restore cilia function are under investigation with potential for success [161].

## 7. Conclusions and Gaps in Knowledge

Motile ciliopathies have significant impact on the newborn, which are understood through the biology of human development. Much of what is currently known of human motile cilia development is deduced from non-mammalian vertebrate models. There remain many gaps in our understanding:Embryonic nodal signaling is postulated but not fully defined, in particular how signaling and dysfunction in the node translates to specific cardiac laterality defects is unknown.In the respiratory system, it is clear that cilia are required for airway clearance, but the role of cilia in the liquid-filled lung, the effect of cilia-mediated flow on the developing lung, and exactly how motile ciliopathies cause respiratory distress at birth is not known.The role of ventricular motile cilia in the development of neural function, outside of the rare contribution by hydrocephalus, is unknown. Perturbations in CSF flow related to malfunctioning cilia may be insufficient to cause clinically apparent disease but may have consequences yet unknown in the developing brain.All motile ciliopathies are studied through the genetic disease PCD and its models, while very little is known of acquired cilia disease in neonates. The respiratory tract and brain ventricular system continue to play critical roles after development throughout adult life and an acquired ciliopathy may develop without underlying genetic disease.

Continued study of human development and disease in the neonate can provide insight into the biology of motile cilia and improve our management of the newborn with motile ciliopathy.

## Figures and Tables

**Figure 1 cells-11-00125-f001:**
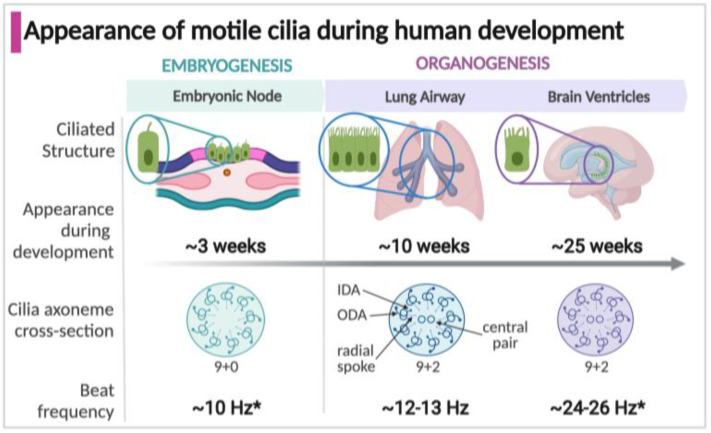
Timeline of the appearance, structure, and function of motile cilia in specialized structures during development. * Some estimates of cilia beat frequency are based on measurements in murine models; Arrows indicate structural details on the ciliary axoneme cross-section (IDA, inner dynein arm; ODA, outer dynein arm).

**Figure 2 cells-11-00125-f002:**
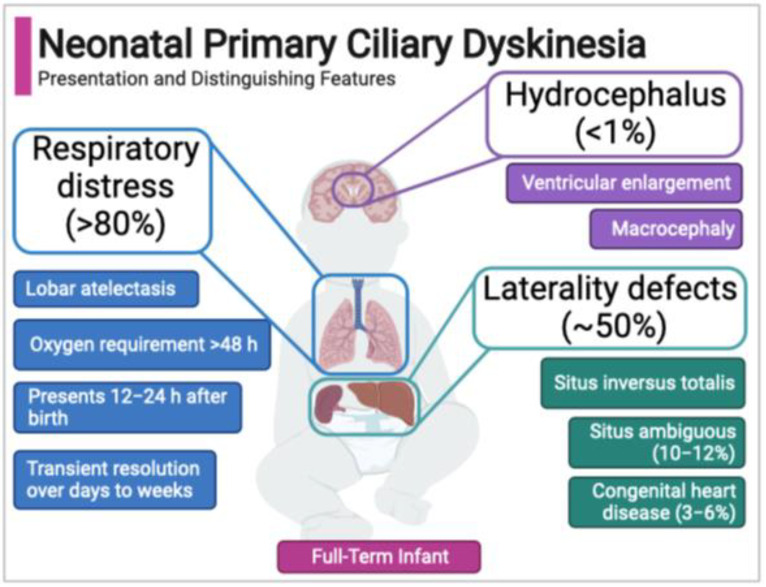
Neonatal features of primary ciliary dyskinesia.

**Table 1 cells-11-00125-t001:** Developmental roles of motile cilia and clinical impact of ciliopathies in the neonate.

Motile Cilia-Containing Structures during Development	Developmental Function	Clinical Complication of Motile Ciliopathies in the Newborn
Embryonic node *Nodal pit cells*	Laterality determination Cardiac formation Visceral organ position	Situs anomalies Congenital heart disease
Respiratory tract *Airway epithelial cells*	Directed airway fluid movement	Respiratory distress of the newborn Lung atelectasis
Brain ventricles *Ependymal cells*	Directed cerebral spinal fluid movement	Ventricular enlargement Hydrocephalus

**Table 2 cells-11-00125-t002:** Selected PCD pathogenic variants by class with reported neonatal presentation.

Functional Class and Gene	Percent of PCD Caused by the Pathogenic Variant	Neonatal Respiratory Distress	Laterality Defect	Hydrocephalus	Ref
**Outer dynein arm**					
*ARMC4 (ODAD2)*	<3%	Yes	Yes	NR	[35,36]
*CCDC103*	<4%	Yes	Yes	NR	[36,37]
*CCDC151 (ODAD3)*	<3%	Yes	Yes	Yes	[38]
*DNAH1*	<1%	Yes	Yes	NR	[36]
*DNAH5*	15–29%	Yes	Yes	NR	[36,37,39,40]
*DNAH11*	6–9%	Yes	Yes	NR	[37,40,41]
*DNAI1*	2–10%	Yes	Yes	NR	[36,39,42]
*TTC25*	<1%	Yes	Yes	NR	[36]
**Dynein axonemal assembly factor**					
*DNAAF2 (KTU)*	<1%	Yes	Yes	NR	[37,43,44]
*DNAAF3*	<1%	Yes	Yes	NR	[39,40]
*DNAAF4 (DYX1C1)*	<1%	Yes	Yes	NR	[39,45]
*LRRC6 (DNAAAF11)*	<1%	Yes	Yes	NR	[37]
*SPAG1*	<4%	Yes	Yes	NR	[37,46]
*ZMYND10*	<2–4%	Yes	Yes	NR	[47]
**Ruler protein**					
*CCDC39*	4–9%	Yes	Yes	NR	[36,37,48]
*CCDC40*	3–4%	Yes	Yes	NR	[36,37,48]
**Central Pair**					
*HYDIN*	<1%	Yes	NR	NR	[36,40,49]
**Radial Spoke**					
*RSPH1*	<1%	Yes (rare)	NR	NR	[50,51]
*RSPH4A*	<1%	Yes	NR	NR	[36]
**Multiciliogenesis**					
*CCNO*	<1%	Yes	NR	Yes	[36,37,52]
*FOXJ1*	<1%	Yes	Yes	Yes	[53,54]
*MCIDAS*	<1%	Yes	NR	Yes	[55]

Abbreviations: Ref, reference.

**Table 3 cells-11-00125-t003:** Indications for diagnostic testing of primary ciliary dyskinesia (PCD) in the newborn.

**Criteria for Neonatal Evaluation of PCD**
Term newborn with >48 h unexplained lung atelectasis (particularly with lobar collapse) or neonatal respiratory distress ^1^ AND 2. A laterality defect OR 3. Family history of PCD
**Diagnostic Testing for PCD in the Neonatal Period**
Genetic diagnosis with pathogenic variants in a known PCD causative gene Structural ciliary defect visualized on transmission electron microscopy ^2^ Ciliary functional abnormalities seen on repeated measures with high-speed video microscopy (beat frequency and/or waveform) ^2^

^1^ Term defined as greater or equal to 37 weeks’ gestation. Neonatal respiratory distress defined by the need for supplemental oxygen or positive pressure ventilation for >48 h without alternative explanation. ^2^ Transmission electron microscopy (TEM) to evaluate cilia ultrastructure and high-speed video microscopy (HSVM) to evaluate cilia beat frequency and wave form, only available at specialized PCD centers.

**Table 4 cells-11-00125-t004:** Initial diagnostic evaluation of the newborn with suspected primary ciliary dyskinesia.

System	Workup
Respiratory	Chest X-ray
Cardiac	Echocardiogram
Gastrointestinal	Abdominal ultrasound (with attention to visceral organ laterality, splenic abnormalities)
Neurologic	Occipital frontal head circumference (OFC) Head ultrasound
Genetic	Genetic testing for cystic fibrosis, immunodeficiency, surfactant deficiency Comprehensive PCD genetic panel

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
