# Peer review of "Impact of Motile Ciliopathies on Human Development and Clinical Consequences in the Newborn"

_cells, 2021, doi:10.3390/cells11010125_

Round 1

Reviewer 1 Report

cells-1500405

The manuscript entitled:Impact of motile ciliopathy on human development and clinical consequences in the newborn” by Rachel Hyland and Steven Brody, a well-known expert in the field, focuses on important but somehow less frequently addressed aspects of PCD.

Based on the title and subtitle of the introduction, the Authors intend to describe cilia and the development of the ciliated tissues/organs in humans. For obvious reasons, some data regarding these processes were obtained in animal models. I would suggest stating more clearly which presented data were obtained by analyses of human cells and which are assumptions made based on animal studies. For example, I am not aware of studies showing that human nodal cilia have 9+0 MTs organization. Accordingly, the Authors cite studies in mice. But while reading the manuscript I could assume that 9+0 organization refers to human nodal cilia. By the way, some 9+2 cilia were found in mice node [Caspary et al., 2007, Dev Cell, Odate et al., 2016, https://doi.org/10.1093/jmicro/dfv352] while 9+2 and 9+4 cilia were seen in rabbit [https://doi.org/10.1002/dvdy.20986].

For the reader outside the field or a newcomer to the field, or a student, the basis for the statement that mutation in certain gene causes (or not) a certain PCD phenotype, is not clear. For example, it is not clear why a mutation in MCIDAS is not associated with laterality defects (lines 163-167).  Please consider also providing a short definition of some terms (for example: polysplenia, dextrocardia) for those who are less familiar with human anatomy and medicine.

Some minor issues can be addressed by simply re-writing the sentence or being more specific.

  1. Line 35: “they contain specialized motor proteins to facilitate bending motion…” – the expression “specialized motor proteins” can be misleading for someone outside the field. It is true that these dyneins are specific for the axoneme but still, these are dyneins. Suggest to re-phrase: they contain specialized motor proteins-containing complexes to facilitate bending motion
  2. Line 36: “Motile cilia are limited to few cell types…” but next Authors listed not cell types but entire organs/structures (except sperm cells). Suggestion: e.g., epithelial cells lining respiratory tracts, brain ventricles, oviduct….
  3. Line 49: “Large motor protein complexes (dyneins) are docked at ruler-like 96 nm intervals along..”

Please be more specific, there are 4 ODAs and 7 different IDAs per 96-nm unit. ODAs within the unit are believed to be identical so their “ruler” is shorter. Please re-phrase. Please add also that central microtubules with docked complexes form the central apparatus.

  1. Fig 1 – embryonic node – please cite the source for this image
  2. Line 69: REF 15 – in this paper, I did not find information that node is present around 3 weeks of gestation in the human embryo. REF 15 states: “nodal cells in the early embryo”. Also, check if REF16 is correct for “lung development at about 10 weeks [16].” and REF 17, 18 for information that” ependymal cilia ….emerge at around 25 weeks”. REF17 – is about cilia in mice, not in humans. I could not check if REF19 and 20 are correct (do not have access),
  3. Subtitle: “Temporal appearance of motile cilia in the human embryo” – suggest to remove word “temporal” – only cilia in two locations described in this part are temporal in the node and esophagus.
  4. Line 83 – the end of the sentence seems to miss something.
  5. Lines 102-103:” …or in ‘ruler’ proteins (CCDC39, CCDC40) which determine placement of IDA and regulatory proteins” – not protein but complexes: regulatory complex (N-DRC) or complexes (radial spokes) that transduce signals from central apparatus
  6. Table 2: please provide the range in % for “rare” description
  7. Table 2: typos: CCDC39 (D is missing)
  8. Lines 116-17: what Authors mean by “limited”:” …precise timing in human embryos is limited) [53]”
  9. Lines 118-119: ” Mouse and zebrafish models show two types of 9+0 mono- ciliated cells within the node”. Mice have also a limited number of 9+2 cilia, while in the case of zebrafish embryos, most cilia in the Kupffer’s vesicle has 9+2 MTs organization (Kramer-Zucker et al., 2005, Tavares et al., 2017)
  10. Line 142: subtitle:” Nodal cilia and the biologic basis of left-right asymmetry” What do Authors mean by the “biological basis”? Maybe simply – L-R asymmetry of the body organs.
  11. Line 233:” cell cycle proteins (e.g., CCNO) amplify centriole numbers” – please re-write, proteins cannot amplify
  12. Lines 240-241:”… which program the production of components required for motile cilia and the motor proteins” – motor proteins as part of the dynein arms are motile cilia complexes (components of motile cilia).
  13. Lines 242-243:” Within the cytoplasm, assembly of the macromolecular dynein arm complexes (for ODA, IDA) is also triggered. Please re-write.
  14. Lines 285-286:” Second, respiratory distress and lobar atelectasis typically occurs 12-24 hours after birth, 285 while the more common respiratory distress syndromes generally present within two 286 hours of birth [102,105]” – this is not quite clear for the reader outside the field which one refers to PCD.
  15. The following publication on ependymal cilia might be of interest.

The adult macaque spinal cord central canal zone contains proliferative cells and closely resembles the human.

Alfaro-Cervello C, Cebrian-Silla A, Soriano-Navarro M, Garcia-Tarraga P, Matías-Guiu J, Gomez-Pinedo U, Molina Aguilar P, Alvarez-Buylla A, Luquin MR, Garcia-Verdugo JM. J Comp Neurol. 2014 Jun 1;522(8):1800-17. doi: 10.1002/cne.23501

Reviewer 2 Report

In this review Hyland and Brody did a good job highlighting the effects of motile cilia disfunction on human development.

I just have a couple of very minor comments regarding some word choices. In this manuscript the authors often use the word "ciliopathy" where I would expect "ciliopathies". E.g. line 15. If the authors really mean one ciliopathy, it would seem better to use "a ciliopathy" or "the ciliopathy PCD" or just "PCD".

In line 82 the authors use "hetergenous" where I think they mean "heterogeneous".

In line 97-98 the authors write "..the result of autosomal recessive genetics". I find the use of "genetics" strange, I would suggest to use "genetic defects" or similar wording.

Author Response

Reviewer 2: In this review Hyland and Brody did a good job highlighting the effects of motile cilia disfunction on human development. I just have a couple of very minor comments regarding some word choices. In this manuscript the authors often use the word "ciliopathy" where I would expect "ciliopathies". E.g. line 15. If the authors really mean one ciliopathy, it would seem better to use "a ciliopathy" or "the ciliopathy PCD" or just "PCD".

Response: We have changed the title “ciliopathy” to “ciliopathies”. We have revised the text to be consistent with the singular and plural use of the term.  

Reviewer 2: In line 82 the authors use "hetergenous" where I think they mean "heterogeneous".

Response: Thank you, we have corrected the text.

Reviewer 2: In line 97-98 the authors write "..the result of autosomal recessive genetics". I find the use of "genetics" strange, I would suggest to use "genetic defects" or similar wording.

Response: The text has been revised to “…the result of autosomal recessive genetic variants, with the exception..”